# Ntrk1 Promotes Resistance to PD-1 Checkpoint Blockade in Mesenchymal Kras/p53 Mutant Lung Cancer

**DOI:** 10.3390/cancers11040462

**Published:** 2019-04-02

**Authors:** Jessica M. Konen, B. Leticia Rodriguez, Jared J. Fradette, Laura Gibson, Denali Davis, Rosalba Minelli, Michael D. Peoples, Jeffrey Kovacs, Alessandro Carugo, Christopher Bristow, Timothy Heffernan, Don L. Gibbons

**Affiliations:** 1Department of Thoracic/Head & Neck Medical Oncology, University of Texas MD Anderson Cancer Center, 1515 Holcombe Blvd, Houston, TX 77030, USA; jmkonen@mdanderson.org (J.M.K.); BLRodriguez@mdanderson.org (B.L.R.); jjfradette@mdanderson.org (J.J.F.); lagibson@mdanderson.org (L.G.); 2Department of Chemistry, Indiana University of Pennsylvania, 1011 South Drive, Indiana, PA 15705, USA; denalihdavis@gmail.com; 3Department of Genomic Medicine, University of Texas MD Anderson Cancer Center, 1515 Holcombe Blvd, Houston, TX 77030, USA; RMinelli@mdanderson.org; 4Center for Co-Clinical Trials, University of Texas MD Anderson Cancer Center, 1515 Holcombe Blvd, Houston, TX 77030, USA; mdpeoples@mdanderson.org (M.D.P.); jjkovacs@mdanderson.org (J.K.); cabristow@mdanderson.org (C.B.); 5Institute for Applied Cancer Science, University of Texas MD Anderson Cancer Center, 1515 Holcombe Blvd, Houston, TX 77030, USA; acarugo@mdanderson.org (A.C.); tpheffernan@mdanderson.org (T.H.); 6Department of Molecular and Cellular Oncology, University of Texas MD Anderson Cancer Center, 1515 Holcombe Blvd, Houston, TX 77030, USA

**Keywords:** non-small cell lung cancer, immunotherapy, PD-1/PD-L1 checkpoint blockade

## Abstract

The implementation of cancer immunotherapeutics for solid tumors including lung cancers has improved clinical outcomes in a small percentage of patients. However, the majority of patients show little to no response or acquire resistance during treatment with checkpoint inhibitors delivered as a monotherapy. Therefore, identifying resistance mechanisms and novel combination therapy approaches is imperative to improve responses to immune checkpoint inhibitors. To address this, we performed an in vivo shRNA dropout screen that focused on genes encoding for FDA-approved drug targets (FDAome). We implanted epithelial and mesenchymal Kras/p53 (KP) mutant murine lung cancer cells expressing the FDAome shRNA library into syngeneic mice treated with an anti-PD-1 antibody. Sequencing for the barcoded shRNAs revealed *Ntrk1* was significantly depleted from mesenchymal tumors challenged with PD-1 blockade, suggesting it provides a survival advantage to tumor cells when under immune system pressure. Our data confirmed Ntrk1 transcript levels are upregulated in tumors treated with PD-1 inhibitors. Additionally, analysis of tumor-infiltrating T cell populations revealed that Ntrk1 can promote CD8+ T cell exhaustion. Lastly, we found that Ntrk1 regulates Jak/Stat signaling to promote expression of PD-L1 on tumor cells. Together, these data suggest that Ntrk1 activates Jak/Stat signaling to regulate expression of immunosuppressive molecules including PD-L1, promoting exhaustion within the tumor microenvironment.

## 1. Introduction

Lung cancer is the leading cause of cancer-related deaths, killing more people in the U.S. than the next three most prevalent cancer types combined [1,2]. The five-year survival rate for all lung cancer patients is about 18%, which has improved marginally over the past several decades even with the improvement of genomic profiling and rational implementation of targeted therapies. Thus, a better understanding of the complexities of lung cancer progression, the contributing microenvironmental factors and how to target them would benefit patient outcomes.

Research focusing on systemic and tumor-infiltrating immune cell populations and their impact on shaping cancer progression in solid tumor types has provided compelling evidence for immune escape as a crucial survival mechanism. These studies have revealed that tumors avoid immune detection through a variety of complex mechanisms. For example, tumors can recruit immunosuppressive populations of cells such as myeloid derived suppressor cells or CD4+ T regulatory cells, which secrete suppressive cytokines that interfere with the cytotoxic functions of CD8+ T cells [3,4,5]. Additionally, tumors upregulate expression of PD-L1, which can occur de novo through oncogenic signaling or as a consequence of IFNγ-stimulation due to immune cell activation. PD-L1 binds to the PD-1 molecule on CD8+ T cells and blocks the full activation necessary for the cytotoxicity [6,7], thus representing an avenue of therapeutic intervention to promote cytotoxic activity of T cells.

The implementation of immunotherapies to release immune system braking mechanisms like those described above has been paradigm-shifting for cancer therapeutics. Clinical studies in lung cancer have revealed that inhibiting the PD-L1/PD-1 axis results in a significantly improved clinical outcome in ~15–20% of patients with lung cancer when compared to standard of care chemotherapy [8,9,10] and thus shows promise in improving patient prognosis. While some patients do show clinical benefit to checkpoint inhibitors when administered as single agents, the majority of patients either show no response or develop resistance to single agent checkpoint inhibition [9,10,11,12,13,14,15]; thus, discovering mechanisms of resistance and tumor cell dependencies in the face of immune-related pressure is imperative in furthering the potential for immunotherapy in treating lung tumors. 

Several factors have been identified as impacting response to immune checkpoint inhibitors. For example, tumor mutational burden significantly correlates with response to immunotherapy, likely due to the creation of neoantigens that activate the immune response. Additional work has focused on oncogenic drivers of lung cancer. Kirsten rat sarcoma (KRAS) mutations occur in about 30% of lung adenocarcinomas, and unlike other common oncogenic drivers (such as epidermal growth factor receptor (EGFR) and anaplastic lymphoma kinase (ALK)), effective targeted therapeutic strategies for KRAS mutant lung cancer have been limited [16]. Interestingly, KRAS mutant lung tumors and their degree of immune system engagement and infiltration vary based upon the co-occurring mutations found within the tumor. Patients that present with a p53 mutation concurrently with oncogenic Kras (KP) exhibit higher expression of PD-L1 and other inflammatory markers when compared to other commonly co-occurring mutations such as STK11/LKB1 or CDKN2A [17], and these patients respond better to PD-1/PD-L1 axis blockade [18]. However, the mechanisms of tumor-regulated immunosuppression and the potential avenues of resistance in KP mutant lung cancer are vastly unknown, and the understanding of these factors is necessary for intelligent use of immunotherapies for maximum benefit to patients.

Previous work in our laboratory has focused on understanding the biology of KP mutant lung tumors through cancer cell intrinsic properties as well as extrinsic factors influencing cancer progression that are present within the tumor microenvironment. We have previously derived murine lung cancer cell lines from the primary or metastatic lesions of the *Kras^LA1/+^/p53^R172HΔg/+^* genetically engineered mouse model of lung cancer [19,20]. These cells demonstrate heterogeneity in their epigenetic state and propensity to metastasize when re-implanted syngeneically into wildtype mice. Specifically, the KP murine cell lines that have undergone an epithelial-to-mesenchymal transition (EMT) are not only more metastatic and aggressive, but they also have lower CD8+ T cell infiltration and an increase in an exhaustive signature when compared to cells in an epithelial state [21]. This heterogeneity also translates to a response to PD-1 blockade, with mesenchymal cells responding initially to the anti-PD-1 antibody but ultimately acquiring resistance [22]. Thus, our in vivo models closely mimic patient disease progression and immune checkpoint inhibitor response, providing the opportunity to discover novel mechanisms regulating tumor response to immune checkpoint blockade in KP mutant lung cancer.

To identify novel mechanisms of KP lung cancer cell resistance to PD-1 checkpoint inhibition, we performed a clinically relevant and powerful in vivo dropout screen. KP murine mouse cell lines stably expressing the FDAome, a library of barcoded shRNAs specific to genes that encode for clinically actionable targets, were implanted into wildtype mice and treated with an anti-PD-1 antibody. Tumors were sequenced and analyzed for depleted shRNA sequences when mice were treated with an anti-PD-1 antibody, thus revealing genes essential for tumor survival in the face of PD-1 blockade. From this screen, neurotrophic receptor tyrosine kinase 1 (Ntrk1) was identified as a top lead candidate as it dropped out significantly in anti-PD-1 treated tumors. Our data indicate that Ntrk1 regulates KP cell biology including cell growth and invasion in vitro while also impacting the tumor-infiltrating immune populations and their functionality with a consistent promotion of an exhausted microenvironment. Thus, we determined that Ntrk1 is a novel regulator of immune functionality in KP lung cancer, and combinatory treatment strategies could circumvent PD-1 blockade resistance.

## 2. Results

### 2.1. An In Vivo Functional Genomics Screen to Identify Novel Tumor Cell Vulnerabilities in the Face of Immune Checkpoint Blockade

To explore novel avenues of therapeutic combinations with immune checkpoint blocking antibodies, we performed a powerful and clinically relevant in vivo dropout screen in combination with PD-1 checkpoint blockade treatment (Figure 1A). The screen library contained short hairpin RNAs (shRNAs) designed against ~200 genes, each of which encoded for a clinically actionable target, termed the FDAome. To ensure robustness and prevent false hits due to shRNA off-target effects, each gene was targeted with 10 unique shRNA sequences. Lentiviral particles expressing the shRNAs were used to transduce two murine Kras/p53 (KP) mutant lung cancer cells. The 393P epithelial cells are a non-metastatic line, whereas the 344P mesenchymal line is an aggressive and metastatic cell line, and each were originally derived from Kras^G12D/+^/p53^R172HΔg^ primary lung tumors as previously described by our laboratory [19]. The 393P and 344P cells stably expressing the FDAome library were implanted subcutaneously into 129/sv wildtype mice (3 mice/treatment group) (Figure 1B). Once tumors reached 150–200 mm^3^, they were then treated with either an isotype control antibody or a PD-1 blocking antibody. 344P tumors, which responded to PD-1 treatment initially but eventually demonstrated resistance (Appendix A), were collected at two time points of anti-PD-1 treatment to identify genes that synergize to prevent the development of resistance. After tumor collection and deep sequencing, quality control measures were completed to ensure sufficient barcode coverage across the library was maintained in vivo (Figure 1C). Importantly, strong separation of hairpins targeting positive controls (Psma1 and Rpl30) and hairpins targeting Luc was observed (Figure 1D, Appendix A). Furthermore, an additional positive control, the proteasomal gene Psmb1, ranks in the top 10 percent of the most significantly depleted genes across all conditions, thus strengthening the validity of the screen hits (Appendix A). To prioritize hits from the screen, a redundant shRNA activity (RSA) score method was used to assign significance of shRNA dropout, then assigning a rank from 1 to 192 for most significant to least significant dropout in each condition.

### 2.2. Short Hairpin RNAs Targeting Ntrk1 Dropped out Significantly from 344P Mesenchymal Tumors Treated with PD-1 Blocking Antibody

The results of the FDAome screen revealed several shRNAs that dropped out from tumors treated with anti-PD-1 treatment, suggesting these genes to be vital for the survival of tumor cells when challenged with immune-related pressure through PD-1 treatment. We compared the differential in RSA value between isotype treated and anti-PD-1 treated tumors to compare nonessential to essential changes in the gene dropout score. Using this metric, we identified Ntrk1 as being one of the hits with the largest differential in RSA value between vehicle and PD-1 treated 344P tumors at both time points of treatment (Figure 2A, Appendix A). This was unique to the mesenchymal 344P tumors, as the epithelial 393P tumors do not show Ntrk1 shRNA dropout in anti-PD-1 treated tumors (Appendix A). A similar screen using the same FDAome library but completed in vitro and in immunocompromised mice did not show significant dropout of Ntrk1 shRNAs [23]. These data suggest that Ntrk1 is nonessential for the survival of tumor cells in normal in vitro and in vivo conditions. However, when challenged with a PD-1 blocking antibody and therefore under immune system pressure, Ntrk1 then becomes essential for tumor cell survival. Importantly, Ntrk1 is a top hit across both time points of tumor collection and sequencing, suggesting the validity of this gene as a positive hit from the screen that is more likely to play a role in PD-1 treatment resistance in the face of long-term treatments.

### 2.3. Transcript Level and Protein Activity of Ntrk1 Are Increased in PD-1 Treated Tumors

Because Ntrk1 dropped out significantly in the FDAome hairpin screen with a PD-1 blocking antibody, we next wanted to determine if the levels of Ntrk1 expression are elevated in tumors treated with immune checkpoint inhibitors. 393P and 344P tumors treated with an IgG control or anti-PD-1 antibody were collected for RNA after 14 or 28 days of treatment. Quantitative real-time PCR (qPCR) revealed that Ntrk1 transcript levels were significantly enhanced in two of the 393P tumors after 14 days of treatment, whereas the 344P tumors showed a more consistent and significant upregulation of Ntrk1 at both time points of treatment (Figure 2B,C). Additionally, another mesenchymal KP murine cell line, 344SQ, was used to derive primary lines from tumors after the development of resistance to PD-1 treatment, which we have previously shown to typically occur between weeks 5 and 7 of treatment [22]. When re-implanted into wildtype mice, these cells showed no response to anti-PD-1 treatment (Appendix A). Compared to isotype treated 344SQ tumors, three of four independent 344SQ PD-1 resistant cell lines showed upregulation of Ntrk1 transcript levels, as well as increased phosphorylation of TrkA protein as shown by western blot (Figure 2D,E). Specifically, the PD-1 resistant cells showed increased expression of the fully glycosylated, mature 140 kDa species of phospho-TrkA, suggesting that downstream signaling of TrkA is more active in these cells. Lastly, 393P and 344P cells were cultured alone or co-cultured with total splenocytes in vitro and treated with an IgG or a PD-1 blocking antibody. Ntrk1 was again found to be significantly upregulated in 344P cells co-cultured with splenocytes that were treated with a PD-1 blocking antibody compared to the isotype control (Figure 2F).

Taken together, these data indicate that anti-PD-1 treatment in tumors or in cell lines co-cultured with an immune compartment upregulates Ntrk1 transcription and activation status, suggesting this pathway may be aberrantly activated as a result of an activated T cell response.

### 2.4. Baseline Expression of Ntrk1 Is Higher in Mesenchymal Murine and Human Lung Cancer Cell Lines and Is Necessary for Invasion and Migration

Previously, our lab generated a panel of KP murine cell lines and profiled them based on their epithelial and mesenchymal status [19,20]. Using these lines, we assayed baseline Ntrk1 expression and found that Ntrk1 levels and phosphorylation correlate with cells in a mesenchymal state (Figure 3A,B), which may explain the differential findings between 393P and 344P from the screen. Similarly, a small panel of human cells delineated by epithelial or mesenchymal status showed the same trend, with Ntrk1 expression correlating with a mesenchymal status. Additionally, cells driven into a mesenchymal state via Zeb1 induction or a more epithelial state with miR-200 induction, as described previously [21,24,25], also confirmed that Ntrk1 expression is higher in cells pushed into a mesenchymal state (Appendix A). Because Ntrk1 expression is higher in mesenchymal cells, we generated stable shRNA-mediated knockdowns in the mesenchymal 344P and 344SQ murine lines and assayed the effect of Ntrk1 knockdown on the ability of these cells to migrate and invade a microenvironment. We found that Ntrk1 knockdown significantly reduced the ability of 344SQ cells to migrate and invade using transwell assays (Figure 3C). Additionally, when plated on a 50% Matrigel:50% collagen type I matrix, the ability of multicellular aggregates to invade was also significantly reduced with Ntrk1 depletion (Figure 3D). Conversely, we generated cell lines stably overexpressing the human cDNA of Ntrk1 in the 393P epithelial murine cell line. Compared to vector control cells, the overexpression of Ntrk1 was sufficient to stimulate invasion and migration (Figure 3E,F). We also analyzed whether another member of the same family, Ntrk3, can impact KP cancer cell biology in a similar manner as Ntrk1. We generated cell lines overexpressing the human cDNA of Ntrk3 in the epithelial 393P cells and determined that these also demonstrate increased invasion and migration (Appendix A). These data indicate that both Ntrk1 and Ntrk3 can regulate cell migration and invasion in KP mutant lung cancer cells, and the expression of Ntrk1 correlates with a more aggressive, mesenchymal state.

### 2.5. Ntrk1 Activates AKT and MAPK Signaling in KP Lung Cancer Cell Lines to Regulate Cell Growth

Additional analysis of Ntrk1-modulated cells revealed that Ntrk1 regulates not only invasion and migration but also cell growth. Specifically, knockdown of Ntrk1 results in a significant reduction of cell growth over time, whereas overexpression of Ntrk1—or exogenous addition of nerve growth factor (NGF) to stimulate TrkA signaling—can stimulate cell growth in epithelial 393P cells (Appendix A).

To determine if Ntrk1 regulates known downstream signaling pathways involved in cell growth and survival, we performed western analysis of Ntrk1 knockdown cell lines. Compared to scrambled control cells, depletion of Ntrk1 decreased both protein kinase B (PKB/AKT) and extracellular receptor kinase 1/2 (ERK1/2) phosphorylation (Figure 3G). Similarly, we utilized the Ntrk1 and Ntrk3 overexpression cells and stimulated signaling with exogenous NGF or NT-3, respectively. In both Ntrk1 and Ntrk3 overexpression cells, AKT and ERK1/2 were quickly and robustly activated, with long term kinetics compared to the control cell line (Figure 3H and Appendix A). We also investigated endogenous signaling cascades regulated by TrkA in KP lung cancer cells. In the 393P and 344P cells stimulated with an NGF ligand, phospho-TrkA was higher at baseline in the mesenchymal 344P cells compared to the 393P cells, as observed in Figure 3B (Appendix A). Additionally, AKT and ERK1/2 signaling cascades were activated in both cell lines, though the kinetics and degree of response differed between them (Appendix A).

To determine if either AKT or MAPK signaling regulates the cell growth phenotype observed with Ntrk1 overexpression, we utilized either MK2206, an AKT specific inhibitor, or trametinib, a mitogen-activated protein kinase kinase (MAP2K/MEK) specific inhibitor. Utilizing MTT viability assays, we determined that while AKT inhibition does inhibit the viability of Ntrk1 overexpression cells, exogenous stimulation of TrkA signaling via NGF was sufficient to circumvent this repression (Appendix A), suggesting that an independent pathway regulates cell growth. By contrast, the addition of trametinib significantly inhibited cell viability, and this was not able to be rescued via exogenous NGF addition.

Taken together, our data indicate that Ntrk1 and Ntrk3 both regulate downstream signaling to AKT and MAPK in KP lung cancer cells, but the regulation of cell growth by Ntrk1 mainly occurs via MAPK signaling cascades.

### 2.6. Ntrk1 Overexpression Promotes Tumor Growth In Vivo and Augments the Tumor Infiltrating Immune Compartment by Promoting CD8+ T Cell Exhaustion

Because Ntrk1 was originally identified from the FDAome hairpin screen as being necessary for tumor cell survival when under immune-mediated pressure via PD-1 blockade, we wanted to determine if Ntrk1 modulates the immune compartment within tumors. To address this, we implanted the Ntrk1 overexpressing and control cells subcutaneously into 129/sv wildtype mice to assay the effects on the immune microenvironment. The overexpression of Ntrk1 significantly enhanced tumor size in vivo (Figure 4A), corroborating the in vitro growth data. Flow cytometry data collected from four-week tumors revealed that Ntrk1 overexpression correlated with a significant reduction of the total T cell infiltrate within the primary tumor, likely due to the significant reduction in total CD8+ T cells, with no significant effect observed on the CD4+ population (Figure 4B–D). Additionally, we found that Ntrk1 overexpression also impacted the functionality of CD8+ cells, with the Ntrk1-expressing tumors presenting with an almost three-fold increase in the level of PD-1+ CD8+ T cells (Figure 4E,F). These cells were also double positive for Tim3, suggesting this population of CD8+ cells were exhausted. In vitro co-culture assays also confirmed that Ntrk1 overexpression can reduce the proliferative capabilities of immune cells as measured by flow cytometry, whereas depleting Ntrk1 promotes the proliferation of immune cells (Appendix A).

These data suggest that Ntrk1 expression promotes tumor growth at least in part by promoting T cell exhaustion.

### 2.7. Combination of Ntrk1 Knockdown with PD-1 Blockade Significantly Reduces Tumor Growth and CD8+ T Cell Exhaustion In Vivo

To further explore the function of Ntrk1 in immunosuppression, we subcutaneously implanted 344P Ntrk1 knockdown or scrambled control cells and treated them either with an IgG control or a PD-1 blocking antibody. The depletion of Ntrk1 alone was sufficient to reduce tumor burden, with the addition of anti-PD-1 treatment further repressing tumor growth (Figure 4G). Tumors were analyzed for infiltrating immune subpopulations using flow cytometry, and while knockdown had no impact on total levels of the immune populations (Figure 4H–J), we found a trend towards decreased CD8+ exhaustion with Ntrk1 knockdown alone, and this was further enhanced with the addition of an anti-PD-1 antibody (Figure 4K,L). Additionally, the population of CD62L+/CD44^high^ effector CD8+ T cells was also significantly increased (Figure 4M), so that the proportion of active CD8 to exhausted CD8 was significantly altered with PD-1 blockade in Ntrk1 depleted tumors. Together with the overexpression analyses, the data demonstrate that Ntrk1 modulates immune functionality with a consistent impact on CD8+ activity and exhaustion.

### 2.8. Ntrk1 Expression Impacts Jak Signaling and Correlates with PD-L1 Levels in Cells Co-Cultured with an Immune Compartment and in Tumor Samples

To determine how Ntrk1 may be modulating tumor growth and the immune microenvironment, we examined Jak/Stat signaling as a function of Ntrk1 expression; this is a known pathway utilized by tumors to promote immunosuppression. By western blot, we found that Ntrk1 knockdown in 344SQ mesenchymal cells did result in decreased phosphorylation of Jak1 and Stat3 as compared to scrambled control cells (Figure 5A). Additionally, stimulation of TrkA signaling via exogenous NGF also increased phospho-Jak1 levels consistently in the 393P cells, whereas phospho-Jak1 levels were high in the 344P cells initially and then fluctuated throughout the time course with NGF stimulation (Figure 5B). One of the well-studied immunosuppressive molecules altered downstream of Jak signaling is Irf1, which in turn triggers the transcription of CD274 or PD-L1. Thus, we assayed whether Ntrk1 can directly impact the expression of PD-L1. In co-culture assays with Ntrk1 knockdown cells, we found that at baseline, Irf1 and PD-L1 expression levels are lower in knockdown versus control cells and this becomes more drastic when knockdown cells are co-cultured with splenocytes (Figure 5C). Conversely, we found a robust and significant increase in Irf1 and PD-L1 expression in Ntrk1 overexpressing cells when cultured with splenocytes (Figure 5D). The upregulation of PD-L1 by Ntrk1 has functional consequences on T cell proliferation. As demonstrated previously, Ntrk1 expression can decrease immune cell proliferation. However, the addition of an anti-PD-L1 antibody to a co-culture of Ntrk1 overexpressing cells with splenocytes can restore immune cell proliferative capabilities (Appendix A). Ntrk1 overexpressing cells also demonstrate a robust increase of PD-L1 expression at 5:1 and 20:1 ratios of splenocytes to tumor cells at the protein level (Figure 5E). Importantly, this upregulation of PD-L1 could be partially reversed when cells were treated with LOXO-101, a pan-Trk inhibitor. These findings were consistent in Ntrk1 overexpressing tumors, with increased PD-L1 levels when compared to vector control tumors (Figure 5F). As expected, treatment with Ruxolitinib, a Jak1/2 inhibitor, significantly inhibited the upregulation of both Irf1 and PD-L1 as a consequence of co-culture with the immune compartment in both vector and Ntrk1-overexpressing cells (Appendix A), suggesting that Trk signaling is just one upstream molecule that impacts Jak-dependent upregulation of PD-L1.

Interestingly, although signaling cascades and biological phenotypes such as invasion and migration are similar between Ntrk1 and Ntrk3 expressing cells, the overexpression of Ntrk3 in 393P cells has no impact on PD-L1 expression when in co-culture with splenocytes (Appendix A). Thus, the regulation of PD-L1 and other immunosuppressive molecules downstream of Jak/Stat signaling cascades may be a unique function of Ntrk1 in KP mutant lung cancer.

Together, these data demonstrate that Ntrk1 expression can modulate KP lung cancer biology as well as the immune microenvironment via Jak1 signaling to promote the expression of immunosuppressive molecules including PD-L1.

## 3. Discussion

The data generated from the in vivo FDAome dropout screen provide compelling evidence for the use of in vivo functional genomic screens to identify novel tumor cell genes and/or pathways that promote immunosuppression and thus may contribute to immunotherapy resistance. We and others have demonstrated the utility of these screens, in the context of immunotherapy or in the context of other tumor biology hallmark dependencies (i.e., cellular growth), to provide preliminary evidence for novel drug targets as well as stimulate new research questions. For example, human patient-derived xenografts and genetically engineered mouse models have been utilized to perform loss-of-function screens using a shRNA library targeting known epigenetic regulators in pancreatic cancer to identify novel tumor survival dependencies [26]. This group provided strong evidence for *WDR5* as being essential for pancreatic tumorigenesis and thus targeting it as a potential therapeutic strategy to further explore for pancreatic cancer patients.

Recently, functional genomic screens have moved towards the CRISPR-Cas9 system for gene editing as it has several advantages over shRNA-based screens, including complete gene knockout as well as greater genomic coverage with larger libraries. Two recent screens did so in the context of tumor cell-immune cell interactions. One was completed in vitro with human T cells in co-culture with melanoma cells to identify tumor genes essential for the effector functions of T cells [27]. The other CRISPR-Cas9 screen was performed in vivo with anti-PD-1 treatment in combination with GM-CSF-secreting, irradiated tumor cell vaccine (GVAX) [28]. This screen contained ~2500 genes and identified hits in the IFNγ response and antigen presentation pathways as expected, but also less understood hits such as *PTPN2*. From these two studies alone, the breadth of knowledge and the resources available to develop hypothesis-driven research about tumor cell influence on immune system response was significantly expanded and will continue to provide additional knowledge about these complex processes.

To contribute to these efforts and better understand response and resistance to immune checkpoint inhibitors in a complex system, we performed a clinically relevant FDAome in vivo dropout screen using a Kras/p53 mutant syngeneic mouse model of lung cancer. This model and the GEM model from which the KP mutant cell lines were originally derived [19] closely recapitulate the progression of human lung cancer disease, with development of metastatic lesions throughout the body, as well as heterogeneity of immune infiltrate and response to immunotherapy agents [19,20,21,22,29]; thus, the similarity to *KRAS*-driven human lung cancer validates the use of these models to address specific immune-related questions, as well as to perform mechanistic and therapeutic studies. We identified Ntrk1 as a top lead hit as being essential for tumor cell survival in vivo when challenged with an anti-PD-1 antibody and therefore a potential avenue of acquired resistance. Molecular studies revealed that Ntrk1 regulates KP lung cancer cell intrinsic biological processes such as cell signaling to AKT and MAPK to promote cellular growth as well as regulation of in vitro invasive capacity. In vivo analyses demonstrated that Ntrk1 also augments immune infiltrate and functionality. Specifically, Ntrk1 expression can promote CD8+ T cell exhaustion within the tumor microenvironment, suggesting that its expression may contribute to CD8+ T cell dysfunction and thus diminish response to PD-1 inhibition.

We now provide the first evidence that Ntrk1 can regulate the expression of the immunosuppressive molecule PD-L1, likely due to modulation of Jak/Stat signaling cascade. In melanoma, loss-of-function *JAK1/JAK2* mutations were discovered in a minority of patients after relapse on the anti-PD-1 treatment pembrolizumab; therefore, it is a potential avenue of acquired resistance to immune checkpoint blockade [30]. Our data may indicate a distinct mechanism by which this pathway becomes aberrantly hyperactivated in lung cancer cells to promote immunosuppression and resistance to immune checkpoint inhibition. Ntrk1 can promote activation of Jak1 and Stat3, and Ntrk1 depletion reduces this signaling and downstream PD-L1 expression. There is little evidence in the literature connecting Ntrk1 to Jak signaling, so this mechanism needs to be further explored. However, previous work in neuronal cells demonstrated that neurotrophin-dependent stimulation of downstream transcription and neuronal cell elongation can be blocked by depletion of Stat3 [31], suggesting that Stat3 does function downstream of neurotrophin receptors. Additionally, knockout of gp130, a type I cytokine receptor, can diminish NGF-induced neurite extension, thus linking Ntrk1 signaling and cytokine signaling. However, both of these studies were limited to neuronal cells and lacked further mechanistic studies to determine whether Ntrk1 directly interacts with these cytokine response elements. Thus, the mechanism for Ntrk1-dependent activation of Jak/Stat signaling remains to be fully elucidated. Additionally, other immunosuppressive molecules that are upregulated as a result of Ntrk1 overexpression in addition to PD-L1 need to be explored to understand the full impact of Ntrk1 upregulation on various immune subpopulations and their functionality within tumors.

In vivo functional genomics screens to address specific immune-related questions such as undiscovered mechanisms of acquired resistance to the anti-PD-1 antibody will drive the field forward and bolster our understanding of the regulatory pathways driving tumor cell evasion of immune detection and death. The goal of the work described was to identify novel therapeutic combinations to improve patient response to immunotherapy. Our data indicate that Ntrk1 may be one such hit that could be carried forward clinically to improve patient response to a single agent PD-1 blocking antibody. In some solid tumor types such as lung cancer, Ntrk1 genetic rearrangements occur infrequently (~1–3% of lung adenocarcinomas), leading to fusion proteins with constitutive kinase activity. Targeted therapies have recently been FDA approved, with most patients showing durable responses [32,33,34,35,36]. Importantly, our preliminary in vivo screen and supporting data suggest that a significantly broader patient population may benefit from these well-tolerated Trk inhibitors in the context of immune checkpoint blockade.

## 4. Materials and Methods

### 4.1. Cell Culture and Reagents

The human lung cancer cell lines used in these studies were H1299, H157, A549, H441, H358, and HCC827. Murine lung cancer cells were created from *Kras^LA1/+^/p53^R172HΔg/+^* genetically engineered mice as previously described [19]. All lung cancer cell lines were cultured in Roswell Park Memorial Institute (RPMI) + 10% Fetal Bovine Serum (FBS). A total of 293T cells were cultured in Dulbecco’s Modified Eagle’s Medium (DMEM) + 10% FBS and were used to generate lentiviral particles for creating stable cell lines. The miR-200ab inducible H1299 cells and Zeb1-inducible 393P cells were generated using the pTRIPz plasmid as previously described by our laboratory [21,24]. Expression of mir-200ab or Zeb1 was induced using 2 μg/mL of doxycycline. Ntrk1 overexpression cells were generated by subcloning human Ntrk1 cDNA from pCMV5 TrkA (Addgene plasmid #15002; http://n2t.net/addgene:15002; RRID:Addgene_15002 [37]) into the pLenti-puro vector backbone using EcoRI and AgeI restriction cut sites. Human Ntrk3 cDNA was cloned into the pLD6E2F vector. The Ntrk1 shRNA sequences that were used in these studies were as follows: sh#1: 5′-TCAAGCGCCAGGACATCATT; sh#2: 5′-GTGGCTGCTGGTATGGTATATCT; sh#3: 5′-TCTATAGCACAGACTATTACC; sh#4: 5′-TTGGAGTCTGCGCTGACTAAT. 

NGF and NT-3 ligands were obtained from Sigma (St. Louis, MO, USA) and used at a final concentration of 100 ng/mL. MK2206, LOXO-101, trametinib, and ruxolitinib inhibitors were obtained from SelleckChem (Houston, TX, USA). Anti-PD-L1 (clone 10F.9G2), PD-1 (clone RMP1-14), and isotype control antibodies (Rat IgG2b and IgG2a, respectively) were obtained from BioXCell (West Lebanon, NH, USA).

### 4.2. Animal Studies

Cancer cells were prepared at a concentration of 1 × 10^6^ cells in 100μl of serum free media. The cells were subcutaneously implanted into the right flanks of male and female syngeneic 129/sv mice of at least three months of age. Tumors were allowed to grow for 3–4 weeks, depending on the study. Where indicated, mice were treated with an anti-PD-1 or isotype control antibody via i.p. injections biweekly (100 μL per dose for a total of 200 μg). After euthanasia, tumors were measured both by calipers and weight and subsequently collected for flow cytometry analyses or sequencing for the FDAome screen (see below). All animal experiments were reviewed and approved by the Institutional Animal Care and Use Committee at the University of Texas MD Anderson Cancer Center (protocol #00001271).

### 4.3. FDAome Dropout Screen

Murine lung cancer cell lines (393P and 344P) were infected at a multiplicity-of-infection (MOI) of 0.3 with a pooled shRNA lentiviral library targeting 192 genes associated with FDA-approved target therapies (10 independent shRNAs/gene). The FDAome expressing cells were implanted into 129/Sv mice at 1.0 × 10^6^ cells/mouse in triplicate for each condition. Once tumor size reached ~150 mm^3^ as measured by calipers, mice were either treated with an isotype control or PD-1 blocking antibody as described above. Tumors were collected after 10 days (time point 1) or 17 days (time point 2) in the 344P model, or after 25 days in the 393P model.

The shRNA-coupled barcodes were detected deploying high-throughput sequencing technology (for detailed procedures and primer sequences see the following reference) [26]. Raw counts for the screen endpoints and a reference population, isolated after transduction, were normalized using the variance stabilizing transformation in R (version 3.3.2) with the DESeq2 in R. A fold change in barcode abundance was estimated by dividing the normalized counts by the reference. Four independent shRNA targeting essential genes (Rpl30, Psma1) or the negative control luciferase (LUC) were cloned with five unique barcodes each and incorporated in the library as positive and negative controls (20 reagents/control, see Appendix A). One LUC hairpin showed an apparent off-target effect, whereas one hairpin for Psma1 did not show a robust drop out; however, this result was not reflective of poor screen performance as the trend was consistent across the five barcodes. After excluding those hairpins, the separation of positive and negative controls was evaluated by the robust strictly standardized mean (SSM, Appendix A). Fold change distribution was converted to percentiles, and biological replicates were collapsed for RSA analysis. The RSA logP-values and ranks are provided in Appendix A.

### 4.4. Cell Viability and Growth Assays

Cell growth in Ntrk1 knockdown or overexpression cells was measured by plating an equal cell number at day 0 and then counting viable cells (using Trypan blue exclusion) every day over a three-day period. Cell growth in Ntrk3 overexpression cells was measured in 3D cultures. These cells were plated on top of a thick Matrigel layer in 8 well chamber slides as single cells. NT-3 ligand was added at the time of cell seeding. Growth was monitored over six days and media refreshed every other day. Images from day 6 were used to measure 3D structure diameter using ImageJ software (version 1.5h). Two independent chambers were quantified for each condition.

Cell viability in the presence of either dimethyl sulfoxide (DMSO,) MK2206 (1 μM), or trametinib (1 μM) was measured using a 3-(4,5-Dimethylthiazol-2-yl)-2,5-Diphenyltetrazolium Bromide (MTT) reagent (Sigma, St. Louis, MO, USA). Cells were plated in a 96 well plate and the drug was added at the time of seeding at the indicated concentrations. After 72 h, an MTT reagent (1.5 mg/mL) was added to each well, incubated for one hour, then supernatant aspirated and precipitate solubilized in DMSO. Absorbance values were read using an Epoch plate reader at 570 nm and 630 nm (background), and background absorbance values were subtracted from 570 nm values.

### 4.5. Invasion and Migration Assays

Cells were plated in an equal number in 8 μm Transwell inserts (BD Biosciences) placed in 24 well plates as described previously [19]. Cells were incubated overnight for 16 h and then stained with a crystal violet solution. Chambers were then imaged by brightfield microscopy on an Olympus IX73 (Olympus, Center Valley, PA, USA), and ImageJ software was used to count cells that had migrated/invaded through the insert pores.

3D cultures were completed as previously described [19]. Briefly, either 100% Matrigel or a Matrigel/collagen type I mixture (50:50) were used to coat 8 well chamber slides. Collagen was used at a final concentration of 1.5–2.0 mg/mL. Single tumor cells were plated on top of the matrix (1500 cells/chamber) in media containing 2% Matrigel. Cells then grew over a period of 3–5 days and invasive structures were manually counted as a percentage of total structures in each well. 30–50 structures were counted per well, and each condition was plated in triplicate.

### 4.6. Co-Culture Assays

Spleens were extracted from 129/Sv mice bearing either 393P or 344P tumors. These were then mechanically processed and filtered to obtain single cells. Red blood cells were lysed using red blood cell (RBC) Lysis buffer (BioLegend, San Diego, CA, USA). Splenocytes were frozen in 90% FBS/10% DSMO. After thaw, viable splenocytes were counted using Trypan blue exclusion and incubated with a far red proliferation dye (Life Technologies, Carlsbad, CA, USA). After 30 min at 37 °C, the dye was washed out with complete media, and stained splenocytes were plated at various ratios with matched tumor cells (i.e., splenocytes from 393P tumors were plated with 393P tumor cells in co-culture) in media supplemented with 5 μg/mL of anti-CD-3 and anti-CD28 (Thermo). Where indicated, an anti-PD-L1 antibody was added at the time of seeding at a concentration of 20 μg/mL. The percentage of far red positive splenocytes was then measured by flow cytometry using a FACSCanto II machine (BD Biosciences, San Jose, CA, USA).

### 4.7. Flow Cytometry Analysis for Immune Subpopulations

Tumors were processed for flow cytometry into single cells using mechanical and enzymatic digestion (enzyme mixture—collagenase, DNAse, and hyaluronidase). Red blood cells were lysed as described above and viable tumor cells counted. The following antibodies were used to stain immune cell populations: Ghost violet 510 Live/Dead, Pacific Blue CD45, PE-594 CD3, PE/Cy7 CD8, APC/Cy7 CD4, BV605 PD-1, APC Tim3, FITC CD62L, BV711 CD44 (BioLegend).

Samples were run on an LSR Fortessa machine. Single color compensation controls were performed using compensation beads (Thermo Fisher Scientific, Waltham, MA, USA) to correct for overlap in signal among antibodies. Spleen samples were used to set a gating strategy for CD3+/CD4+ and CD3+/CD8+ T cells. FlowJo software (version 10) was used to perform all downstream analyses on subpopulations.

### 4.8. Western Blot Analysis

Cells were harvested and lysed in radioimmunoprecipitation assay buffer (RIPA) lysis buffer supplemented with phenylmethylsulfonyl fluoride (PMSF), a protease inhibitor (Cell Signaling), and phosphatase inhibitors (Sigma). Lysates were separated by SDS-PAGE (BioRad, Hercules, CA, USA), transferred to nitrocellulose or polyvinylidene fluoride (PVDF) (BioRad) membranes, and probed with the following primary antibodies: phospho-TrkA(Y674/675), phospho-TrkA(Y785), TrkA, phospho-MEK, MEK, phospho-ERK, ERK, phospho-AKT, AKT, phospho-GSK3b, phospho-Jak1, Jak1, phospho-Stat3, Stat3 (Cell Signaling, Danvers, MA, USA), phospho-TrkC, total TrkC (Thermo Fisher), PD-L1 (Abcam, Cambridge, MA, USA), and actin (ProteinTech, Rosemont, IL, USA).

### 4.9. RNA Extraction and Real-Time qPCR

RNA was extracted from cells in vitro using a TRIzol reagent (Thermo Fisher). Tumor cell RNA was extracted using the mirVana RNA extraction kit (Life Technologies). Briefly, tumors were collected and snap frozen, then processed in 300 μL of lysis buffer using homogenization. RNA was then extracted as directed by the kit protocol.

All RNA samples were quantified, and reverse transcription was performed with 2 μg of RNA using qSCRIPT cDNA SuperMix (Quantabio, Beverly, MA, USA). Real-time PCR was performed using primer sets specific for each gene (obtained from Origene, Rockland, MD, USA) and the SYBR^®^ Green PCR Master Mix (Life Technologies). L32 (60S ribosomal gene) was used to normalize expression across samples.

### 4.10. Statistical Analyses

All analyses were performed using GraphPad Prism software (version 7.01). Unpaired Student’s *t* tests were used in comparison of two conditions, and one-way ANOVA was used for comparisons of three or more conditions. Tukey’s test was used to correct for multiple comparisons. All analyses were 2-tailed and *p*-values < 0.05 were regarded as significant.

## 5. Conclusions

Functional genomics screening in vivo generates novel scientific questions in the context of a complex ecosystem within a whole animal. The FDAome in vivo dropout screen revealed Ntrk1 as a novel regulatory molecule of both KP tumor cell intrinsic biology and extrinsic immune microenvironment factors. Our data indicate that Ntrk1 regulates KP tumor cell invasion, migration, as well as signaling downstream to survival and growth pathways. Additionally, Ntrk1 augments the immune profile within tumors, pushing them towards a dysfunctional, exhausted state. Together, these data suggest that targeted Trk inhibitors may have utility in combination therapy strategies with immune checkpoint inhibitors to improve lung cancer patient response and survival. 

## Figures and Tables

**Figure 1 cancers-11-00462-f001:**
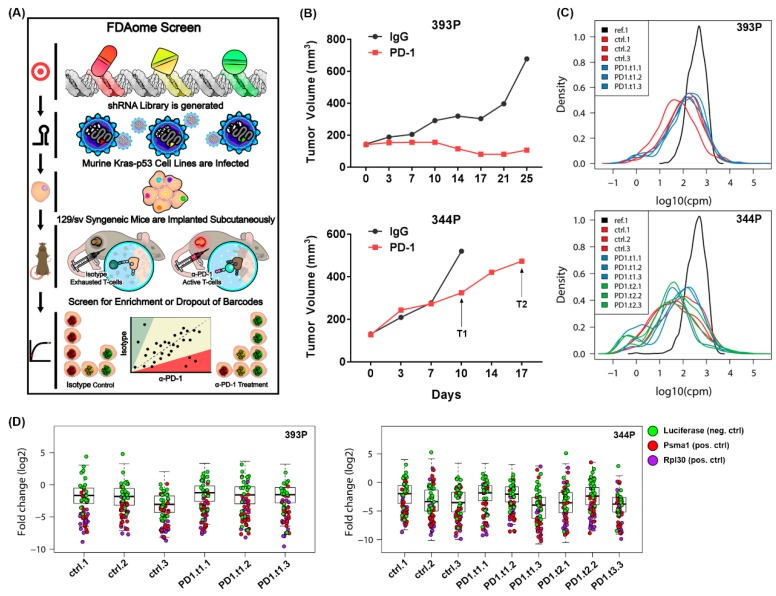
An in vivo functional genomics screen to identify novel tumor survival vulnerabilities when treated with PD-1 blocking antibody. (**A**) Schematic illustrating the workflow of the FDAome short hairpin RNA (shRNA) dropout screen. Briefly, a library of lentiviral particles expressing 10 different barcoded shRNAs for each of 192 genes was transduced into murine Kras/p53 mutant lung cancer cells. Genes included in the library have FDA-approved drugs that target the gene product. These cells were implanted into syngeneic 129/Sv and tumors were treated with PD-1 blocking antibody or IgG control. Tumors were sequenced for barcoded shRNAs and compared to reference cells for enriched or depleted shRNAs between isotype and PD-1 treated tumors. (**B**) 393P epithelial cells and 344P mesenchymal cells used in the FDAome screen were implanted subcutaneously and tumor growth was measured via calipers. T1 indicates time point 1 of tumor collection with PD-1 treatment in the 344P tumors, and T2 indicates time point 2. (**C**) Viral integration distribution of reference population and tumors determined through barcode sequencing (cmp counts per million). Ctrl.1-3 indicate 3 independent IgG treated tumors, and PD1.t1.1-3 label triplicates of anti-PD-1 treated tumors collected at the early time point. PD1.t2.1-3 indicate 344P tumors which were collected 1 week later at time point 2. (**D**) Fold change distribution (log2) of individual tumors relative to the initial transduction references with individual barcodes for positive controls Psma1 (red)/Rpl30 (purple), and the negative control Luc (green). Individual tumor samples per each group (ctrl, PD1.t1, PD1.t2) are shown.

**Figure 2 cancers-11-00462-f002:**
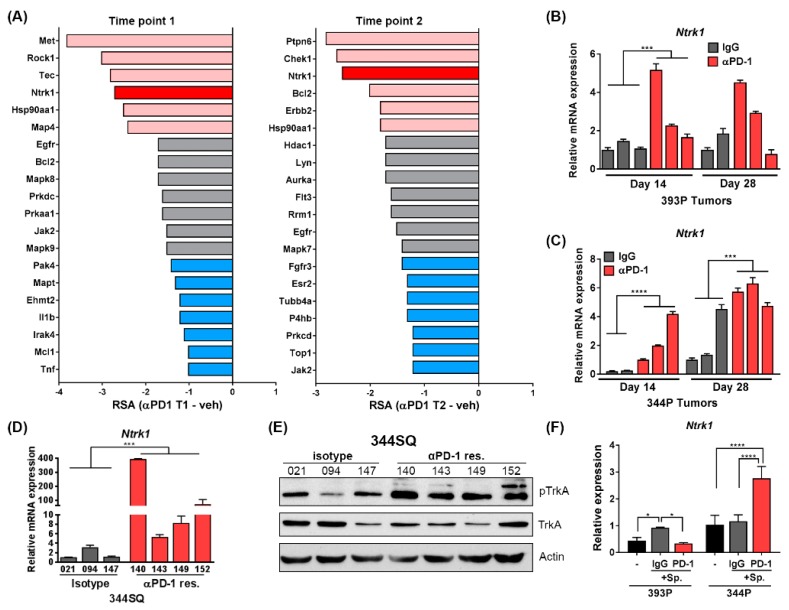
Ntrk1 shRNA drops out significantly from 344P tumors treated with anti-PD-1 and is upregulated in tumor cells treated with anti-PD-1 antibody. (**A**) Results from FDAome shRNA dropout screen in 344P tumors graphed as a differential score. The RSA from the isotype treatment condition for each gene was subtracted from the RSA from the same gene in the anti-PD-1 treatment group. Time point 1 is shown to the left, and time point 2 to the right. The top 10% of hits are depicted and these are divided into three groups, with the most significant shown in red. Ntrk1 is highlighted in bold red in both graphs. (**B**) Real-time PCR (qPCR) data for Ntrk1 expression in 393P tumors treated with anti-PD-1 or isotype control for 14 or 28 days. Expression values are normalized against L32 reference gene, relative to one isotype treated tumor sample. (**C**) Ntrk1 expression was analyzed in 344P tumors as described in panel B. (**D**) 344SQ tumor cells implanted in syngeneic mice were treated with isotype control or anti-PD-1 antibody. Tumors were collected after development of resistance to PD-1 blockade (~week 6), cell lines were derived from the tumors, and Ntrk1 expression analyzed via qPCR. The numbers below bars denote the individual mouse from which the cell line was derived. (**E**) Cell lines described in panel D were probed for phospho-TrkAY674/675 levels by Western blotting. (**F**) 393P and 344P cells were cultured in vitro either alone (-) or in the presence of splenocytes (sp.) at a 5:1 ratio of splenocytes to tumor cells. The treatment of either IgG or PD-1 blocking antibody was added to the co-culture and after 4 days, splenocytes were washed out and tumor cells were collected for RNA. Ntrk1 expression was analyzed using qPCR.

**Figure 3 cancers-11-00462-f003:**
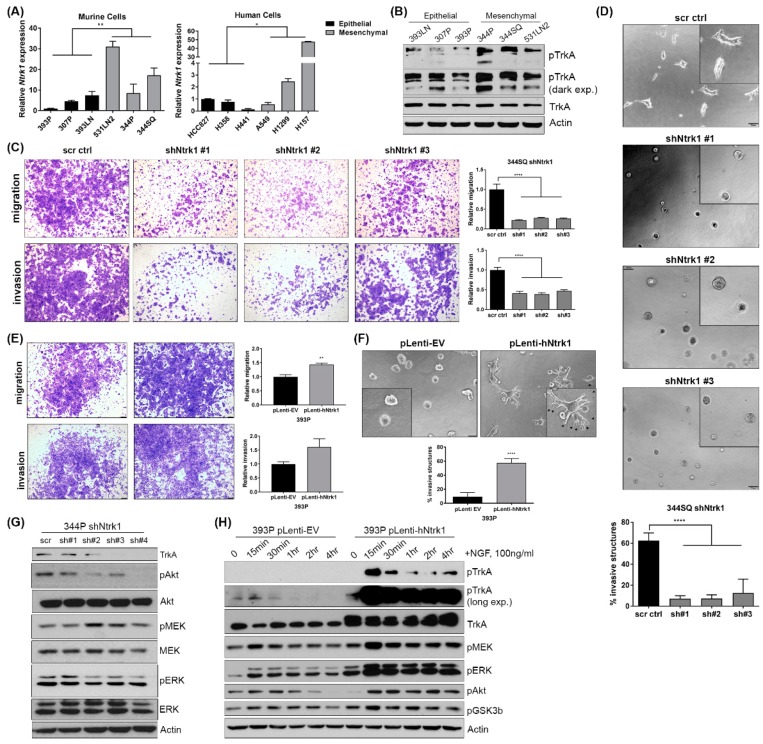
Ntrk1 expression correlates with cells in a mesenchymal state and regulates cell migration, invasion, and AKT and MAPK signaling pathways. (**A**) Real-time qPCR data for Ntrk1 expression in 3 epithelial and 3 mesenchymal murine KP lung cancer cell lines (left) and in 3 epithelial and 3 mesenchymal human lung cancer cell lines (right). (**B**) Western blot showing phospho-TrkA expression in murine KP lung cancer cells as a function of epithelial and mesenchymal status. Actin was used as a loading control. (**C**) Ntrk1 was stably depleted using 3 different shRNA sequences. 344SQ Ntrk1 knockdown cells were plated in triplicate in transwell chambers with Matrigel to measure invasion or without to measure migration. After 16 hours, cells that had migrated or invaded were fixed and stained with crystal violet. Representative images of a single chamber are shown. Quantifications were done using ImageJ and are graphed to the right as a fold change compared to non-targeting scrambled control cells. (**D**) 344SQ shNtrk1 cells were plated on a 50% Matrigel:50% collagen type I matrix. Images were taken after 3 days and the percentage of invasive structures was quantified and graphed below. (**E**) Human cDNA encoding for Ntrk1 was stably introduced into 393P cells. These cells were plated in migration and invasion transwell assays as described in panel C. Representative images are shown and quantifications are graphed to the right. (**F**) 393P Ntrk1 overexpressing cells were plated on a Matrigel and collagen type 1 mixed matrix as described in panel D. Invasive structures were quantified and are shown in the graph below. (**G**) Western blot analysis of 344P Ntrk1 knockdown cells to examine protein expression of TrkA to confirm knockdown and signaling pathways including AKT and MEK/ERK. Actin was used as a loading control. (**H**) 393P empty vector (EV) and Ntrk1 overexpression cells were stimulated with NGF over the course of 4 hours and protein collected for western blot analysis of phospho-TrkA, and AKT and MEK/ERK signaling pathways.

**Figure 4 cancers-11-00462-f004:**
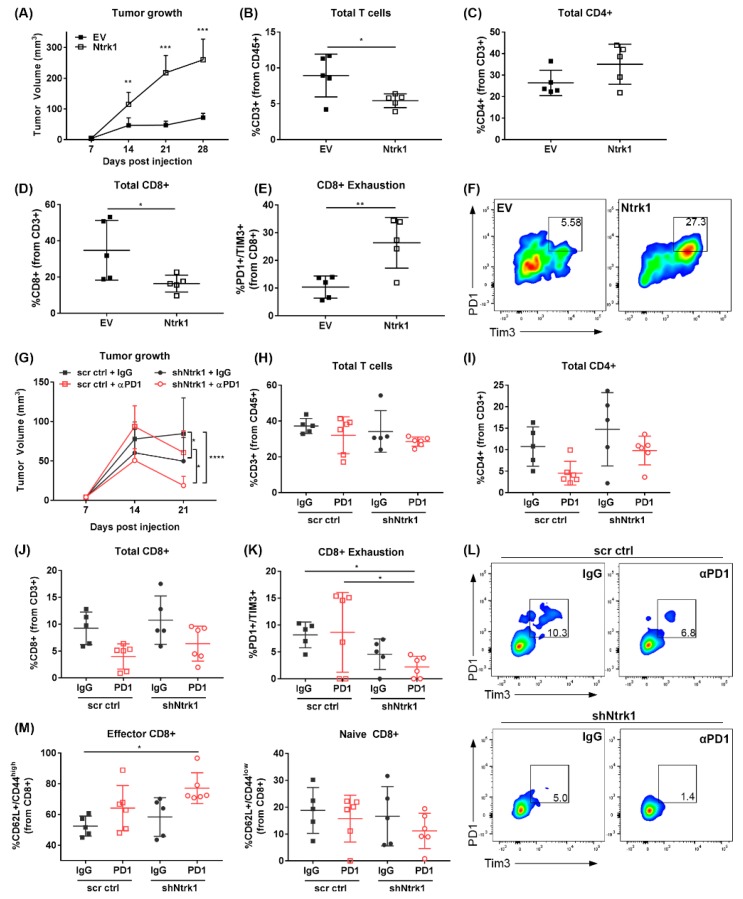
Ntrk1 modulates tumor growth as well as CD8+ T cell exhaustion and activity in vivo. (**A**–**F**) 393P EV and Ntrk1 cells were implanted subcutaneously into 129/sv wildtype mice. After 4 weeks of growth, tumors were collected and processed for flow cytometry analysis of immune cell populations. *n* = 5 mice/cell line. (**A**) Tumor growth was monitored weekly via caliper measurements. (**B**) The percentage of CD3+ T cells in the microenvironment was gated from live CD45+ cells. (**C**) CD4+ T cells were calculated as a percentage of total CD3+. (**D**) CD8+ T cells were calculated as percentage of CD3+ T cells. (**E**) T cell exhaustion was measured as PD-1+/TIM3+ from the CD8+ population. (**F**) A representative dot plot of PD1+/TIM3+ T cells from each condition. (**G**–**M**) 344P cells with stable depletion of Ntrk1 were implanted subcutaneously into 129/sv wildtype mice and compared to a non-targeting control cell line. One week post-implantation, tumors were then treated with either IgG control or PD-1 blocking antibody for 2 weeks, at which point tumors were collected and processed for flow cytometry analysis. (**G**) Tumor growth was monitored weekly via caliper measurements. (**H**) The total CD3+ T cells was quantified from total live CD45+ cells in tumors. (**I**) CD4+ T cells were calculated from total CD3+ cells. (**J**) CD8+ T cells were quantified as a percentage of total CD3+ T cells. (**K**) PD-1+/TIM3+ exhausted T cells were calculated as described in panel E. (**L**) Representative dot plots from each condition showing PD1+/TIM3+ T cells. (**M**) Effector (left graph) and naïve (right graph) CD8+ T cells were calculated as a percentage from total CD8+ T cells. Effector cells were characterized as CD62L+/CD44high and naïve cells as CD62L+/CD44low.

**Figure 5 cancers-11-00462-f005:**
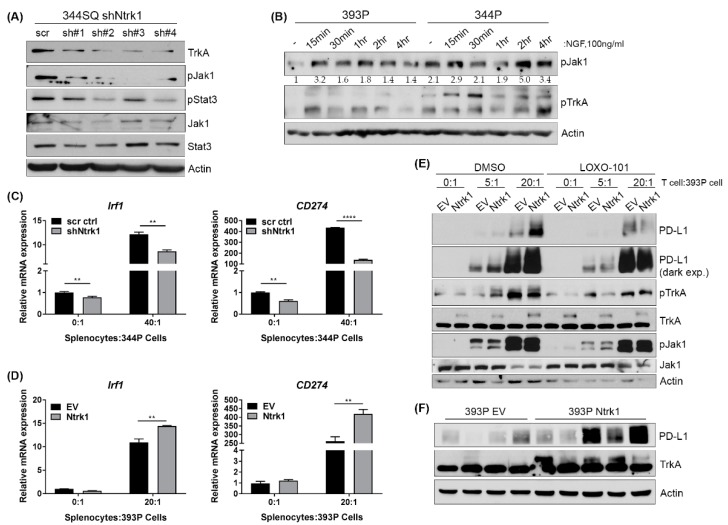
Ntrk1 can regulate JAK/STAT signaling to promote expression of PD-L1. (**A**) Western blot analysis of 344SQ Ntrk1 knockdown cells probing for phospho-Jak1 and phospho-Stat3. Actin was used as a loading control. (**B**) Western blot analysis of 393P and 344P cell lines stimulated with NGF at 100ng/ml over a short time course to assay activation of phospho-Jak1 and phospho-TrkA. Densitometry calculations of phospho-Jak1 intensity relative to actin were normalized to 393P (-) and are shown below the phospho-Jak1 blot. (**C**) Real-time qPCR analysis of CD274 (PD-L1) expression in 344P Ntrk1 knockdown cells when co-cultured with total splenocytes. (**D**) 393P Ntrk1 overexpression cells were co-cultured with total splenocytes and assayed using qPCR for expression of CD274. (**E**) 393P Ntrk1 overexpression cells were cultured alone (0:1) or co-cultured with total splenocytes at 5:1 and 20:1 ratios. Either DMSO control or LOXO-101, a pan Trk-inhibitor, were added to the co-culture at the time of seeding. After 3 days, tumor cells were assayed for protein expression of PD-L1, phospho-TrkA, and phospho-Jak1. (**F**) Western blot analysis of PD-L1 expression in 393P empty vector (EV) control and Ntrk1 overexpression tumors.

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
