# Peer review of "Ntrk1 Promotes Resistance to PD-1 Checkpoint Blockade in Mesenchymal Kras/p53 Mutant Lung Cancer"

_cancers, 2019, doi:10.3390/cancers11040462_

Reviewer 1 Report

Overall, this is an interesting work studying the popular target in cancer field. I believe it will be of interest to the readers of the journal. But there are quite a number of mistakes and mis-labeling in the manuscript, and some of the experimental settings are not clearly demonstrated. The results of some figures are not consistent with the context. Therefore, it needs to be carefully revised before can be accepted for publication.

1) In the Figure 1 C and Figure 1D, what are the meaning of the titles shown in graph, such as ctrl1, PD1.t1.1, etc.

2) The number of mice and group division in the experiment of in vivo functional genomics screen is not clear. More details should be provided.

3) In the Fig 2B and Fig 2C, the numbers of sample group are not consistent among different groups, please explain the reason and provide more details. Besides that, in the Fig 2D and Fig 2E, suppose the samples used in figure 2D and 2E are from the same batch of samples, but the experiment outcome is not consistent, even the sample number cannot match up with each other. Please provide a reasonable explanation.

4) As is mentioned in the Result Part 2.1, “Once tumors reached 150-200mm3, they were then treated with either an isotype control antibody or a PD-1 blocking antibody.” While in the later experiment, as is mentioned in the Figure 4G, tumors were treated with antibody one-week post-injection. What is the rational of choosing different timepoints of drug treatment. And this may make it hard to investigate the effect of antibody.

5) As is mentioned in the Result Part 2.8, Fig 5B showed that stimulation of TrkA signaling via exogenous NGF can also increase phospho-Jak1 levels. But from Figure 5B, we can’t get this conclusion.

6)The current research of 2.8 is not sufficient enough to support their hypothesis. Firstly, did they consider the potential effect of JAK stimulation on regulating Irf1 in the cell lines used in the experiment showed in Fig5? and in Figure 5 e, is it possible to add inhibitor targeting ERK, AKT, and JAK to see the correlation of Nrkt, related signaling pathway and PD-L1 expression.

Author Response

        1. In the Figure 1 C and Figure 1D, what are the meaning of the titles shown in graph, such as ctrl1,         PD1.t1.1, etc.

We agree that these labels were missing from the figure and figure legend and appreciate the Reviewer raising this to our attention. We have updated both the figure and legend to improve the clarity. The labels in Figures 1C,D indicate the group name (ctrl refers to isotype treated samples, PD1 refers to PD1 blocking antibody treated samples). The number after the group name indicates independent tumor samples that were processed and sequenced.

2. The number of mice and group division in the experiment of in vivo functional genomics screen is not clear. More details should be provided.

To clarify this point we have now included additional experimental details of the in vivo functional genomics screen, including the number of mice per group, in the results and methods sections of the text (lines 119; 393). Also, the manuscript which first utilized the in vitro functional genomics screen methodology has now been published, which includes extensive additional information about the methodology that was also used for this combination screen with PD-1 blockade (DH Peng, et al. ZEB1 suppression sensitizes KRAS mutant cancers to MEK inhibition by an IL17RD-dependent mechanism. Sci. Transl. Med. 11, eaaq1238 (2019)). This citation was updated in the text as well (line 143).

 3. In the Fig 2B and Fig 2C, the numbers of sample group are not consistent among different groups, please explain the reason and provide more details. Besides that, in the Fig 2D and Fig 2E, suppose the samples used in figure 2D and 2E are from the same batch of samples, but the experiment outcome is not consistent, even the sample number cannot match up with each other. Please provide a reasonable explanation.

For Fig 2B and 2C, we only have RNA from limited numbers of tumor samples that have been treated with PD-1 blocking antibody, so 2-3 independent samples were used per treatment arm. We understand the confusion about Figures 2D and 2E. Thus, we have now revised the figure to ensure the samples in the qPCR graph are displayed in the same order as shown on the western. We have updated these figures with the individual mouse number as well to decrease confusion and improve the clarity.

 4. As is mentioned in the Result Part 2.1, “Once tumors reached 150-200mm3, they were then treated with either an isotype control antibody or a PD-1 blocking antibody.” While in the later experiment, as is mentioned in the Figure 4G, tumors were treated with antibody one-week post-injection. What is the rational of choosing different timepoints of drug treatment. And this may make it hard to investigate the effect of antibody.

For the treatment experiment outlined in Figure 4, our primary goal was to determine the direct impact of the anti-PD-1 antibody treatment in Ntrk1 knockdown tumors on the presence and functionality of immune cell subpopulations. To do so, we wanted to use an early time point, to make a connection between the change in immune cell populations and effect on tumor growth. Also, to obtain clean data output from the flow cytometric analysis of tumors, it is essential to ensure that tumors are not so large as to become necrotic or ulcerated, which requires an early time point of collection. We have also previously published data illustrating the use of anti-PD-1 antibody administered starting at this time point (Chen L, et al. Cancer Discovery, 2018). Lastly, for the in vivo screen, it was essential to have sufficient tumor tissue to perform downstream sequencing, thus requiring a slightly larger tumor mass at the initiation of treatment.

 5. As is mentioned in the Result Part 2.8, Fig 5B showed that stimulation of TrkA signaling via exogenous NGF can also increase phospho-Jak1 levels. But from Figure 5B, we can’t get this conclusion.

The data in Figure 5B demonstrate that exogenous stimulation of NGF increased pJak1 levels throughout all time points of collection in the 393P cells. The 344P cells have higher pJak1 at baseline, and upon exogenous NGF stimulation we observe an increase early but with fluctuations over the time course. We have now updated the results section to objectively comment on these results of Fig 5B (lines 262-264), and we have updated the figure to include densitometry measurements of pJak1 relative to the loading control under each respective lane.

 6. The current research of 2.8 is not sufficient enough to support their hypothesis. Firstly, did they consider the potential effect of JAK stimulation on regulating Irf1 in the cell lines used in the experiment showed in Fig5? and in Figure 5 e, is it possible to add inhibitor targeting ERK, AKT, and JAK to see the correlation of Nrkt, related signaling pathway and PD-L1 expression.

We appreciate the constructive feedback regarding the data in Figure 5. With regards to Jak stimulation of Irf1 in our cell lines, we have updated the western blot in Fig 5E to include phospho-Jak1, which demonstrates that co-culturing the 393P cells (including the empty vector cells) with total splenocytes is sufficient to stimulate Jak1 signaling. Likewise, the qPCR data shows a robust increase in Irf1 in the 393P empty vector cells in co-culture in the same conditions as utilized in the western, suggesting that the presence of the immune compartment does stimulate Jak signaling in the tumor cells and downstream Irf1 (Fig 5C,D). However, Ntrk1 overexpression significantly increases the degree to which Irf1 and PD-L1 are upregulated in the co-culture conditions, whereas knockdown decreases this effect. Thus, these data suggest that Ntrk1, at least in part, can regulate Jak signaling and the downstream expression of Irf1 and PD-L1.

We agree that assaying the impact of Jak inhibition on PD-L1 expression in our system would be interesting, especially downstream of Ntrk1. Thus, we have included data in Fig S6 that demonstrates the repression of Irf1 by qPCR and PD-L1 by western blot with Ruxolitinib treatment (JAK inhibitor). We also added the description of this figure to the text (lines 279-283). With the limited time provided for the revision process, we were unable to address the question about the impact of targeting ERK and AKT on PD-L1 in Ntrk1 overexpressing cells, but agree this will be an interesting question to address in the future.

Reviewer 2 Report

The manuscript “Ntrk1 promotes resistance to PD-1 checkpoint blockade in Kras/p53 mutant lung cancer” is novel, relevant and very well performed. No doubt recognizing resistance mechanisms and new therapeutic approaches are essential to help immunotherapy. The in vivo shRNA dropout screen is an elegant tool. Focus on Ntrk1 and downstream signaling cascades related to the expression of PD-L1 on tumor cells was a good choice. Authors convincingly show Ntrk1 expression modulates KP lung cancer biology and the immune system via Jak1 signaling, resulting in the expression of immunosuppressive molecules as PD-L1.

Some results focus our attention to the originality and beauty of the work. For instance, Ntrk1 knockdown decreased the ability of 344SQ cells to migrate and invade, and overexpression of Ntrk1 was enough to stimulate invasion and migration. The comparison with 393P tumors was important and relevant for resistance mechanisms studies. Ntrk1 also modulated cell growth and enhanced tumor size in vivo. Ntrk1 ability to promote tumor growth involved T cell exhaustion at least partially. Evidence about AKT and MAPK signaling involvement in Ntrk1 effect on cell growth and the impact on jak signaling and correlation with PD-L1 levels in co-cultures nicely finish the work.

Author Response

The manuscript “Ntrk1 promotes resistance to PD-1 checkpoint blockade in Kras/p53 mutant lung cancer” is novel, relevant and very well performed. No doubt recognizing resistance mechanisms and new therapeutic approaches are essential to help immunotherapy. The in vivo shRNA dropout screen is an elegant tool. Focus on Ntrk1 and downstream signaling cascades related to the expression of PD-L1 on tumor cells was a good choice. Authors convincingly show Ntrk1 expression modulates KP lung cancer biology and the immune system via Jak1 signaling, resulting in the expression of immunosuppressive molecules as PD-L1.

Some results focus our attention to the originality and beauty of the work. For instance, Ntrk1 knockdown decreased the ability of 344SQ cells to migrate and invade, and overexpression of Ntrk1 was enough to stimulate invasion and migration. The comparison with 393P tumors was important and relevant for resistance mechanisms studies. Ntrk1 also modulated cell growth and enhanced tumor size in vivo. Ntrk1 ability to promote tumor growth involved T cell exhaustion at least partially. Evidence about AKT and MAPK signaling involvement in Ntrk1 effect on cell growth and the impact on jak signaling and correlation with PD-L1 levels in co-cultures nicely finish the work.

We would like to thank Reviewer 2 for their positive feedback regarding the work included in this manuscript. We agree that recognizing novel avenues of intrinsic and acquired resistance to immune checkpoint inhibitors such as anti-PD-1 is an important area of research in order to maximize responses to immunotherapy in lung cancer.

Round  2

Reviewer 1 Report

Comments

1. As is mentioned in the Result Part 2.8, “the addition of an anti-PD-L1 274 antibody to a co-culture of Ntrk1 overexpressing cells with splenocytes can restore immune cell proliferative capabilities (Fig S6).(line 273-275)” .This result is not shown in FigS6, but in Fig S7.

2. In the previous experiment, the tumors were treated with anti-PD-1 antibody, and the effect of anti-PD-1 antibody on the 393P tumor, 344P tumors or 344SQ tumors could be found. However, as is mentioned in FigS7, the Ntrk1 overexpressing cells were treated with anti-PD-L1 antibody, what is reason for changing the antibody. What effect does anti-PD-L1 have on the 393P tumor, 344P tumors or 344SQ tumors. Could it inhibit growth of these tumors. If could, Could you add the related result? If not, what could we get from these result?

3. As is mentioned in the Result Part 2.1, “344P tumors, which respond to PD-1 treatment initially but eventually demonstrate resistance [23](line 121-122)”, But from Ref.23, I can’t find any result related with 344P tumors, how could you get this conclusion from this reference? Besides that, from Figure 4G, we could find that in 129/sv wildtype mice injected with control cells, the tumor volume of mice with anti-PD-1 is smaller compared with IgG. Do you have statistic analysis of these two groups to see whether there is a significant difference between these two groups. And could you show how IgG and anti-PD-1 antibody affect tumor growth in 129/sv wildtype mice implanted with the 393P and 344P cells, which were not infected with the FDAome library. And from the existing result, it hardly to conclude that 344P tumors showed the resistance to PD-1 treatment eventually and Ntrk promotes resistance to PD-1 treatment in344P tumors. The result just showed that Ntrk could enhance the effect of PD-1 treatment in 344P tumors.

4. And besides that, the title is “Ntrk1 promotes resistance to PD-1 checkpoint blockade in KRAS/p53 mutant lung cancer”. But as is shown in manuscript, Ntrk1 could not affect the impact of PD-1 treatment on the 393P tumors. So Ntrk could not influence PD-1 treatment in all KRAS/p53 mutant lung cancer. Strictly speaking, Ntrk may just affect PD-1 treatment in KRAS/p53 mutant mesenchymal lung cancer.

5 The result shown in Result Part 2.8 concluded that Ntrk1 expression impacts Jak signaling and correlates with PD-L1 levels in cells co-cultured with an immune compartment and in tumor samples. However, I can’t find the significance of detecting PD-L1 levels in these cells. Have you detected the PD-L1 level in mices mentioned above. Could you find the correlation between PD-L1 level and the effect of PD-1 treatment on tumor in mices mentioned above. If so, it is valuable to investigate the correlation between Ntrk1 and PD-L1 expression and explore related mechanism.

Author Response

1. As is mentioned in the Result Part 2.8, “the addition of an anti-PD-L1 274 antibody to a co-culture of Ntrk1 overexpressing cells with splenocytes can restore immune cell proliferative capabilities (Fig S6).(line 273-275)” .This result is not shown in FigS6, but in Fig S7.

 We appreciate the Reviewer pointing out this oversight. We have now updated the order of the supplemental figures and corrected this error.

2. In the previous experiment, the tumors were treated with anti-PD-1 antibody, and the effect of anti-PD-1 antibody on the 393P tumor, 344P tumors or 344SQ tumors could be found. However, as is mentioned in FigS7, the Ntrk1 overexpressing cells were treated with anti-PD-L1 antibody, what is reason for changing the antibody. What effect does anti-PD-L1 have on the 393P tumor, 344P tumors or 344SQ tumors. Could it inhibit growth of these tumors. If could, Could you add the related result? If not, what could we get from these result?

We utilized anti-PD-L1 in this experiment due to the significant increase in the PD-L1 molecule as a result of Ntrk1 overexpression (Fig 5D-F). Specifically, we wanted to know if PD-L1 upregulation on the tumor cells impacted T cell function, and thus targeting PD-L1 directly was the most direct and logical experiment.

The impact of anti-PD-L1 treatment on these tumors has previously been published by our laboratory (Chen L, et al. Nature Comm. 2014). These data indicate that mesenchymal tumor cells (such as 344P, 344SQ) express high levels of PD-L1 on their cell surface and therefore initially respond to anti-PD-L1. However, similar to anti-PD-1 treatment, these tumors do ultimately acquire resistance (Chen L, et al. Cancer Discovery 2018), so targeting either side of the PD-L1/PD-1 axis leads to similar results. Epithelial cells express significantly lower PD-L1 and therefore do not respond to anti-PD-L1 antibody treatment in vivo (Chen L, et al. Nature Comm. 2014).   

3. As is mentioned in the Result Part 2.1, “344P tumors, which respond to PD-1 treatment initially but eventually demonstrate resistance [23](line 121-122)”, But from Ref.23, I can’t find any result related with 344P tumors, how could you get this conclusion from this reference? Besides that, from Figure 4G, we could find that in 129/sv wildtype mice injected with control cells, the tumor volume of mice with anti-PD-1 is smaller compared with IgG. Do you have statistic analysis of these two groups to see whether there is a significant difference between these two groups. And could you show how IgG and anti-PD-1 antibody affect tumor growth in 129/sv wildtype mice implanted with the 393P and 344P cells, which were not infected with the FDAome library. And from the existing result, it hardly to conclude that 344P tumors showed the resistance to PD-1 treatment eventually and Ntrk promotes resistance to PD-1 treatment in344P tumors. The result just showed that Ntrk could enhance the effect of PD-1 treatment in 344P tumors.

As our laboratory has previously shown, the mesenchymal cell lines demonstrate similar patterns of CD8+ T cell infiltration, exhaustion and functionality, as well as trends in expression of immunosuppressive molecules such as PD-L1 (Chen L, et al. Nature Comm. 2014). We are now including in the supplement growth curves of wildtype tumors from 393P and 344P without the FDAome library, treated with anti-PD-1 (Fig S1). 344P tumors, similar to other mesenchymal lines shown in Ref 23, initially respond to PD-1 blockade but ultimately acquire resistance. Conversely, 393P tumors demonstrate a more durable response to treatment. These results are in line with the FDAome tumor growth curves shown in Fig 1B.

Significance for Fig 4G occurs where indicated between the groups. Thus, while there is a trend with anti-PD-1 in the control cells compared to IgG treatment, this comparison was not significant using 2-way ANOVA and Tukey’s multiple comparisons test.  

4. And besides that, the title is “Ntrk1 promotes resistance to PD-1 checkpoint blockade in KRAS/p53 mutant lung cancer”. But as is shown in manuscript, Ntrk1 could not affect the impact of PD-1 treatment on the 393P tumors. So Ntrk could not influence PD-1 treatment in all KRAS/p53 mutant lung cancer. Strictly speaking, Ntrk may just affect PD-1 treatment in KRAS/p53 mutant mesenchymal lung cancer.

 We appreciate the feedback and now have included “mesenchymal” in the manuscript title.

5. The result shown in Result Part 2.8 concluded that Ntrk1 expression impacts Jak signaling and correlates with PD-L1 levels in cells co-cultured with an immune compartment and in tumor samples. However, I can’t find the significance of detecting PD-L1 levels in these cells. Have you detected the PD-L1 level in mices mentioned above. Could you find the correlation between PD-L1 level and the effect of PD-1 treatment on tumor in mices mentioned above. If so, it is valuable to investigate the correlation between Ntrk1 and PD-L1 expression and explore related mechanism.

We understand the comment regarding the significance of PD-L1 upregulation via Jak signaling on response to PD-1 treatment, and we have not yet explored this in vivo. However, our prior publications clearly demonstrate that PD-L1 expression on tumors cells is the major driver of CD8+ T cell exhaustion in our models, and there is significant evidence in the field that one mechanism of PD-1 resistance is hyperexhaustion of T cells (e.g., Wei F, et al. PNAS 2013; O’Donnell J, et al. Cancer Treatment Reviews 2017). Thus, it is mechanistically reasonable that such significant upregulation of PD-L1, corresponding with a significant increase in exhausted CD8+ T cells, contributes to treatment resistance as a result of Ntrk1 overexpression. Additionally, we acknowledge the need for further work in this area in the discussion (lines 350-353). Here, we comment that examination of other immunosuppressive molecules besides PD-L1 would also be worthwhile, as JAK/STAT signaling can regulate numerous immunosuppressive molecules downstream in addition to PD-L1. We appreciate the Reviewer’s point, as this will be an important broader area for to explore in our future studies.